# Dual-Wavelength Excited Intense Red Upconversion Luminescence from Er^3+^-Sensitized Y_2_O_3_ Nanocrystals Fabricated by Spray Flame Synthesis

**DOI:** 10.3390/nano10081475

**Published:** 2020-07-28

**Authors:** Xiaofan Zhao, Zeyun Wu, Zining Yang, Xu Yang, Yiyang Zhang, Maohui Yuan, Kai Han, Changqing Song, Zongfu Jiang, Hongyan Wang, Shuiqing Li, Xiaojun Xu

**Affiliations:** 1College of Advanced Interdisciplinary Studies, National University of Defense Technology, Changsha 410073, China; zhaoxiaofan11@nudt.edu.cn (X.Z.); diablo_3rd@126.com (Z.Y.); fractal_yangxu@outlook.com (X.Y.); hankai0071@nudt.edu.cn (K.H.); songchangqing08@nudt.edu.cn (C.S.); jiangzongfu7@163.com (Z.J.); xuxj@21.cn.com (X.X.); 2Hunan Provincial Key Laboratory of High Energy Laser Technology, National University of Defense Technology, Changsha 410073, China; 3Department of Energy and Power Engineering, Key Laboratory for Thermal Science and Power Engineering of Ministry of Education, Tsinghua University, Beijing 100084, China; wuzeyun18@mails.tsinghua.edu.cn; 4Key Laboratory of Advanced Reactor Engineering and Safety of Ministry of Education, Collaborative Innovation Center of Advanced Nuclear Energy Technology, Institute of Nuclear and New Energy Technology, Tsinghua University, Beijing 100084, China; zhangyiyang@mail.tsinghua.edu.cn; 5Department of Physics and Chemistry, PLA Army Academy of Special Operations, Guangzhou 510507, China; 6State Key Laboratory of Pulsed Power Laser Technology, National University of Defense Technology, Changsha 410073, China

**Keywords:** multicolor tuning, dual-wavelength excitation, upconversion luminescence, spray flame synthesis, Y_2_O_3_ nanocrystal, biological applications

## Abstract

Er^3+^-sensitized upconversion nanoparticles (UCNPs) have attracted great attention due to their tunable upconversion (UC) emissions, low cytotoxicity, high resistance to photobleaching and especially multiple effective excitation wavelengths. However, detailed energy conversion between Er^3+^ and Tm^3+^ ions in Y_2_O_3_ UCNPs is still a problem, especially under multi-wavelength and variable pulse width excitation. In this work, we successfully fabricated a series of Er^3+^-sensitized Y_2_O_3_ nanocrystals by a spray flame synthesis method with a production rate of 40.5 g h^−1^. The as-prepared UCNPs are a pure cubic phase with a mean size of 14 nm. Excited by both 980 and 808 nm lasers, the tunable upconversion luminescence (UCL) from Er^3+^ ions was achieved by increasing the Er^3+^ doping concentration, co-doping Tm^3+^ ions and extending excitation pulse-width. The investigations of the lifetimes and the laser power dependence of UC emissions further support the proposed mechanism, which provides guidance for achieving effective color control in anticounterfeiting and multiplexed labeling applications. In addition, the red UC emission at about 5 mm beneath the tissue surface was observed in an ex vivo imaging experiment under the excitation of 808 nm laser, indicating that the Y_2_O_3_:Er^3+^/Tm^3+^ UCNPs have great prospects in further biological applications.

## 1. Introduction

Lanthanide-doped upconversion nanoparticles (UCNPs), which can convert a long wavelength excitation into an anti-Stokes emission [1], have shown great advantages in anticounterfeiting, photodynamic therapy and bioimaging [2,3,4,5,6], because of their outstanding properties like tunable upconversion (UC) emissions, low cytotoxicity, large Stokes shifts, high resistance to photobleaching and photochemical degradation [7].

Generally, in order to achieve bright specific upconversion luminescence (UCL) in these UCNPs, the host materials are simultaneously doped with two types of lanthanide ions as a so-called sensitizer and activator. The sensitizer ions absorb the near-infrared light and then transfer the energy to activator ions, which accept the energy and generate visible UCL. The excitation wavelength and UCL efficiency of the UCNPs are greatly dominated by the sensitizer ions [8]. Benefited from the large absorption cross-section at 980 nm and the efficient energy transfer (ET) processes with activator ions, Yb^3+^ ions have become the most commonly used sensitizer [9,10,11]. However, for living tissue applications, due to the high absorption coefficient of water molecules at 980 nm, Yb^3+^-sensitized UCNPs are facing severe challenges of a laser-induced overheating problem and thus a decrease in the depth of penetration [12]. Obviously, this limitation of single efficient excitation wavelength largely impedes the future development of Yb^3+^-sensitized UCNPs.

To meet diverse application needs, great efforts have been done to find new sensitizers with multiple effective excitation wavelengths. With the recent 808 and 980 nm lasers showing efficient excitations, Er^3+^ sensitizers have become promising candidates [13,14,15]. For instance, under the excitation of 980 nm laser, the various emission wavelengths of Er^3+^-sensitized UCNPs provide the ability of multicolor tuning from green to red, which is important for anticounterfeiting and multiplexed labeling applications [4]. Further, under the excitation of 808 nm lasers, Er^3+^-sensitized UCNPs can shift the excitation wavelength from 980 nm to the near infrared (NIR) biological window (NIR-I, 700–900 nm) [16], making them suitable for photodynamic therapy, bioimaging and in vivo biosensing [17,18,19]. In particular, the red UC emission (650 nm) of Er^3+^ locates in the visible biological window (600–700 nm), indicating the superiority of Er^3+^-sensitized UCNPs with intense red UCL in the deep living tissue biological applications [20]. To efficiently obtain intense red UCL in Er^3+^-sensitized UCNPs, Tm^3+^ ions are usually chosen as the co-doped ions to enhance the population of the red-emitting state through building an ET and back energy transfer (BET) route [21,22,23,24].

For the fabrication of co-doped (e.g., Er^3+^/Tm^3+^) UCNPs with high crystallinity and quantum efficiency, liquid-phase synthesis methods (solvothermal and hydrothermal) were demonstrated to be effective [7,25]. However, in liquid-phase synthesis strategies, the governing parameters including shielding gas, reaction time, temperature, pressure and concentration must be controlled strictly and the synthesis process always takes tens of hours with a low production rate [26,27]. It greatly restricts the practical applications of the high quality UCNPs. So far, the continuous large-scale synthesis of ultra-small UCNPs is still a challenge to be solved. It is worth noting that gas-phase synthesis technology (e.g., flame aerosol synthesis) has become a potential approach due to its rapid manufacture, high-throughput production and continuous synthesis capabilities [28,29,30,31,32,33,34,35]. Recently Ju et al. successfully synthesized Yb^3+^/Er^3+^ co-doped UCNPs through flame spray synthesis method [36]. However, there is still no relevant literature on the flame synthesis of Er^3+^-sensitized UCNPs with relatively large production rates. Therefore, both the fundamental and experimental investigations are still needed.

In this paper, we describe a class of Er^3+^-sensitized Y_2_O_3_:Er^3+^/Tm^3+^ UCNPs fabricated by the spray flame synthesis method using a self-built swirl-stabilized spray flame reactor. Under the excitation of both 980 and 808 nm lasers, we investigated the color tuning ability of the as-prepared UCNPs and further discussed the UCL mechanism. Moreover, in order to estimate their applied potential in biological applications, we examined the UCL penetration ability of Y_2_O_3_:Er^3+^/Tm^3+^ UCNPs in fresh pork.

## 2. Materials and Methods

### 2.1. Materials

Y(NO_3_)_3_·6H_2_O (99.9% metals basis), Er(NO_3_)_3_·5H_2_O (99.9% metals basis), Tm(NO_3_)_3_·6H_2_O (99.9% metals basis) and *n*-Butanol (98% CP) were purchased from Aladdin Industrial Corporation (Shanghai, China). 2-Ethylhexanoic acid (99%) was purchased from Alfa (Shanghai, China). All the chemicals were used as received, without further purification.

### 2.2. Synthesis of the Y_2_O_3_:Er^3+^/Tm^3+^ UCNPs

A self-built swirl-stabilized spray flame reactor, as illustrated in Figure 1, was used to fabricate the Y_2_O_3_:Er^3+^/Tm^3+^ UCNPs. This reactor contains two main parts: the swirl-stabilized flame burner and the spray atomizer. The swirl-stabilized flame burner has eight tangential slits, 1 mm in width and 15 mm in length. The fuel (CH_4_, 3 L min^−1^) and the oxidant (Air, 30 L min^−1^) were separately injected into the burner from neighboring slits to prevent flame flashback. Due to the small cross-sectional area of the tangential slit and the large swirl number of burner (32.72), a strong heat recirculating flow of high-temperature burned gas establishes a rapidly mixed combustion zone [37]. Particularly, under the turbulent condition, the stabilization effect becomes more significant and helps to achieve the large production rate of UCNPs [38,39].

For the preparation of the precursor solution, firstly, 7.5 mmol Y(NO_3_)_3_, Er(NO_3_)_3_ and Tm(NO_3_)_3_ were mixed at their respective stoichiometric amounts in a beaker. Then, 22.5 mmol 2-Ethylhexanoic acid (2-EHA) was added to the beaker, ensuring that the molar ratio of Ln^3+^ and 2-EHA is 1:3. By adding 2-EHA, the precursor becomes easily volatile through ligand exchange, and the synthesis process will follow the gas-to-particle route, which favors the uniform small-sized and solid nanoparticles rather than large hollow particles [40]. Then the solution was fixed to 50 mL by adding corresponding volume of n-Butanol. Finally, the precursors were well mixed by an ultrasonic water bath for 1 h.

The spray atomizer has a dual fluid structure, the liquid precursor was injected by a syringe pump (LD-P2020II) at 1200 mL h^−1^ and the atomizing gas (air, 15 L min^−1^) was delivered to shred the liquid precursor into micron droplets. It was installed at the central bottom of burner and sprayed micron aerosols into the rapidly mixed swirl flame from the vertical direction to form a core spray flame. Then, the Y_2_O_3_:Er^3+^/Tm^3+^ UCNPs nucleate and grow in the core spray flame region. A water-cooling plate was fixed above the reactor to collect the UCNPs by thermophoresis, and the as-prepared UCNPs are shown in Appendix A.

### 2.3. Instruments and Measurements

The crystallographic features of the as-prepared UCNPs were examined by an X-ray diffractometer (XRD) with Cu K radiation at 40 kV and 40 mA (Bruke D8 Advance, Hangzhou, China). The morphology and size of the Y_2_O_3_:Er^3+^/Tm^3+^ UCNPs were characterized by transmission electron microscopy (TEM; Tecnai G2 F20, FEI, Changsha, China). The compositional elements of the UCNPs were measured by an energy dispersive spectrometer (EDS). For photoluminescence experiments, the 808 and 980 nm fiber coupled diode lasers (BWT K808DAHFN-25.00W, BWT K976DA3RN-30.00W, Changsha, China) were used as the excitation sources. Additionally, the generated UCL was collected and detected by a monochromator (Zolix Omni-λ300i, Changsha, China) and a photomultiplier (PMT). The photos of the UCL color were taken by a Complementary Metal Oxide Semiconductor (CMOS) sensor (Sony IMX519, Changsha, China). All the above photoluminescence measurements were performed at room temperature, and the test samples were prepared by dissolving the UCNPs in ethanol solution with a mass concentration of 0.5 mg mL^−1^.

## 3. Results and Discussion

### 3.1. Structure and Morphology

The XRD patterns of Y_2_O_3_:Er^3+^/Tm^3+^ UCNPs doped with different Tm^3+^ concentrations are demonstrated in Figure 2a. All the diffraction peaks corresponded to the standard pure cubic-phase Y_2_O_3_ (JCPDS no. 43-1036), and no heterogeneous diffraction peak was observed. The results indicate that the doping of Er^3+^ and Tm^3+^ ions had no significant influence on the phase purity of the Y_2_O_3_ hosts. Additionally, it also revealed that the high-temperature flame environment of this reactor was suitable for forming high crystallinity Y_2_O_3_ nanoparticles.

The morphology images of the single-doped Y_2_O_3_:Er^3+^ and the co-doped Y_2_O_3_:Er^3+^/Tm^3+^ UCNPs are shown in Figure 2b,c. All of them have a rectangle-shape morphology, which agreed well with the cubic crystalline phases. The particle size distribution of these UCNPs was uniform, with a mean size of 14 nm. The compositional elements of the UCNPs were examined by EDS, as shown in Appendix A. The Y, O and Er elements are clearly shown in the EDS analysis result. Additionally, detailed elemental composition of the Y_2_O_3_:Er^3+^ (8 mol%) UCNPs is also shown in Appendix A. It can be found that the doping ratio of the Er^3+^ ions was 9.55 mol%, which is close to the nominal doping concentration (8 mol%). This indicates that the lanthanide ions could be effectively doped into the Y_2_O_3_ hosts through the spray flame synthesis method.

### 3.2. Color Tuning under 980 nm Excitation

As is well-known, the luminescence intensity of Er^3+^-sensitized UCNPs will reduce when the Er^3+^ doping concentration is too high, mainly caused by the cross-relaxations (CRs) between Er^3+^ ions leading to the concentration quenching [41]. In order to determine the inflection point of concentration quenching, we varied the Er^3+^ doping concentrations gradually from 0.5 to 15 mol%. Under the excitation of 980 nm laser, the absolute luminescence spectra of the Y_2_O_3_:Er^3+^ UCNPs doped with different Er^3+^ concentrations are shown in Appendix A. The green (550 nm) and red (650 nm) UC emissions could be clearly observed, which were ascribed to the (^2^H_11/2_, ^4^S_3/2_)→^4^I_15/2_ and ^4^F_9/2_→^4^I_15/2_ transitions from the Er^3+^ ions, respectively. Both the green and red UC emissions will increase when the doping concentrations of Er^3+^ ions rise from 0.5 to 8 mol%. However, when the doping concentration of Er^3+^ ions further increases to 15 mol%, all UC emissions start to decrease gradually due to the significant concentration quenching effect [41,42,43]. Hence, we selected the Er^3+^ doping concentration range from 0 to 8 mol% as the focus of further investigation.

Then, we continued to investigate the influence of Tm^3+^ ions on the color tuning of UC emissions. The UC emission spectra of Y_2_O_3_:Er^3+^/Tm^3+^ UCNPs is shown in Figure 3a. As the Tm^3+^ concentration increased from 0 to 4 mol%, the red-to-green (R/G) UC emission intensity ratio dramatically increased from 1.59 to 12.59. Furthermore, the tendencies of integral UCL intensities and R/G ratios are illustrated in Figure 3b. Obviously, the integral luminescence intensities monotonically decreased with the doping concentrations of Tm^3+^ ions. On the contrary, the R/G ratios continuously increased. When the doping concentration of Tm^3+^ was 1 mol%, the R/G ratio reached 8.5. A further increase of the doping concentration will largely reduce the integral luminescence intensity and the R/G ratio only slightly increases. Therefore, the optimal doping concentration of Tm^3+^ is acceptable when fixed at 1 mol%.

To explore the principle of color tuning more clearly, we fixed the Tm^3+^ doping concentrations at 0 and 1 mol%, and simultaneously varied the Er^3+^ doping concentrations from 0 to 8 mol%. Under the excitation intensity of 341 W cm^−2^, the corresponding UCL spectra and the simplified ET mechanism are displayed in Figure 4. For Er^3+^ single-doped Y_2_O_3_ UCNPs, the UCL color changes from green to yellow as the Er^3+^ doping concentrations increase from 0.5 to 8 mol%, which is shown in the inserts of Figure 4a. Additionally, the R/G ratios rise from 0.09 to 1.59 correspondingly. After co-doping 1 mol% Tm^3+^ ions into the Y_2_O_3_:Er^3+^ UCNPs, the luminescence color can be tuned from green to red, as shown in the inserts of Figure 4b. The R/G ratios increase from 0.55 to 8.50. Figure 4c shows a comparison of the R/G ratios of these Y_2_O_3_:Er^3+^ UCNPs doped with and without Tm^3+^ ions. The R/G ratios of Y_2_O_3_:Er^3+^/Tm^3+^ UCNPs were at least 4.4 times larger than Er^3+^ single-doped UCNPs, indicating that the introduction of Tm^3+^ ions had a strong enhancement on the UCL color tuning ability. To explain the above results, the simplified mechanism for color tuning is illustrated in Figure 4d. When low concentration of Er^3+^ is single-doped, the distance between adjacent Er^3+^ ions is relatively far. It is difficult to transfer energy between each other and thus the UCL color is more inclined to green. When the Er^3+^ doping concentration increased, the average distance between two Er^3+^ ions became smaller, which promoted the CRs. This effect enhanced the population of the red-emitting state and suppressed the population of green-emitting states, inducing the UCL color to turn yellow [41]. However, the CRs between two Er^3+^ ions were reversible and it will limit the increase of the R/G ratio, where it only reached 1.59. When introducing Tm^3+^ ions, besides the CR processes between adjacent Er^3+^ ions, there were also other processes between Er^3+^ and Tm^3+^ ions (ET and BET) that can effectively improve the proportion of red UCL. Notably, based on these processes, the UCL color can be tuned from green to red, which is quite useful for anticounterfeiting and multiplexed labeling applications [4].

To further understand the color tuning mechanism, the possible ET and BET processes, non-radiative transitions and UC emissions are depicted in Figure 5. When the Er^3+^ doping concentration was low, the ground-state absorption (GSA) and the excited-state absorption (ESA) were the main population processes, where the electrons in the ground state of Er^3+^ were excited to ^4^I_11/2_ by GSA, then excited to ^4^F_7/2_ by ESA and finally populated the emitting states (^2^H_11/2_, ^4^S_3/2_ and ^4^F_9/2_) through non-radiative transitions (left part of Figure 5). These processes result in the proportion of green-emitting states being relatively higher than the red one [41], which is consistent with the observed results (Figure 4d).

When the Y_2_O_3_ UCNPs were highly doped with Er^3+^ ions, besides the GSA and ESA processes mentioned above, the adjacent Er^3+^ ions could transfer energy to each other through energy transfer upconversion (ETU) and CR processes (middle part of Figure 5) [41,44,45,46,47]. After absorbing a 980 nm photon, the electrons in the ground state of Er^3+^ will reach the ^4^I_11/2_ state. Except the ESA process, sectional electrons in the ^4^I_11/2_ state reach the ^4^I_13/2_ state by non-radiative transition, and then populate the ^4^F_9/2_ state by an ETU process: ^4^I_11/2_(adjacent Er^3+^) + ^4^I_13/2_(Er^3+^)→^4^I_15/2_(adjacent Er^3+^) + ^4^F_9/2_(Er^3+^) between the adjacent Er^3+^ ions. Other electrons in the ^4^I_11/2_ state are excited to the ^4^F_9/2_ state through the resonant CR transitions of ^4^F_7/2_→^4^F_9/2_ and ^4^I_9/2_→^4^I_13/2_ in adjacent Er^3+^ ions (CR1 and CR3). In addition, the electrons in the ground state of Er^3+^ can be excited to the ^4^I_13/2_ state, corresponding to the resonant CR transition of ^2^H_11/2_, ^4^S_3/2_→^4^I_9/2_ in adjacent Er^3+^ ions (CR2). CR1 and CR2 can reduce electrons pumped to the ^2^H_11/2_ and ^4^S_3/2_ states. CR1 and CR3 can populate the ^4^F_9/2_ state of Er^3+^, and CR2 will increase electrons in ^4^I_13/2_ state of Er^3+^, which can be further pumped to the ^4^F_9/2_ state by ETU. All the ETU and CR processes mentioned above can promote the transfer of electrons from green-emitting states (^2^H_11/2_ and ^4^S_3/2_) to red-emitting state (^4^F_9/2_), which greatly increased the proportion of red UC emission. The energy gaps of CR1, CR2 and CR3 were 151 cm^−1^, 377 cm^−1^ and 742 cm^−1^, respectively, which were comparable with intrinsic phonons of Y_2_O_3_ hosts (597 cm^−1^) [48,49]. It means that phonon assisted CRs can easily occur, and the R/G ratio can be enhanced by simply increasing the Er^3+^ doping concentration.

However, although the color of UCL can be tuned from green to yellow by increasing the Er^3+^ doping concentration, it is still difficult to achieve pure red UC emission only relying on the CRs between the adjacent Er^3+^ ions. Since the CRs between two Er^3+^ ions are reversible and it will limit the increase of R/G ratios. Besides CRs, we can also increase the proportion of red UC emission through ETU process. However, there is a large energy gap between the ^4^I_13/2_ and ^4^I_11/2_ states (3609 cm^−1^), which essentially limits the population of red-emitting state (^4^F_9/2_) through ETU. To solve this problem, Tm^3+^ ions were added into the Y_2_O_3_:Er^3+^ UCNPs, and the possible ET processes are shown in the right part of Figure 5. Here the key energy state involved is ^3^H_5_ of Tm^3+^ ions, which is an intermediate energy state between the ^4^I_13/2_ and ^4^I_11/2_ states of Er^3+^ ions. The electrons in the ^4^I_11/2_ state can easily be transferred to the ^3^H_5_ state by ET process, and then be transferred from the ^3^H_5_ state to the ^4^I_13/2_ state by the BET process. These two processes are more likely to happen because their energy gaps (1829 cm^−1^ and 1780 cm^−1^ respectively) were much smaller than the energy gap between the ^4^I_13/2_ and ^4^I_11/2_ states (3609 cm^−1^). Therefore, the introduction of Tm^3+^ ions can help electrons to be transferred from the ^4^I_11/2_ state to the ^4^I_13/2_ state, which increases the proportion of the red UC emission significantly. However, it is noteworthy that the wavenumber of the ^3^F_4_ state of Tm^3+^ is close to the ^4^I_13/2_ state of Er^3+^, the electrons on the ^4^I_13/2_ state can also be transferred to the ^3^F_4_ state simultaneously. Consequently, the introduction of Tm^3+^ ions will slightly decrease the absolute intensity and lifetime of the red UCL.

To further support the proposed transition mechanism, we measured the time–decay curves and the power dependence lines of the green (564 nm) and red (661 nm) UC emissions, as shown in Figure 6. The results in Figure 6a,b show that both the green and red UCL intensities show exponential decay after being excited by a 980 nm pulsed laser. When the Tm^3+^ doping concentrations increased from 0 to 4 mol%, the lifetimes of 564 nm UC emission decreased from 268.3 to 100.1 µs, and the lifetimes of 661 nm UC emission decreased from 322.7 to 117.3 µs. The detailed lifetime trends of 564 and 661 nm UC emissions as a function of the Tm^3+^ doping concentrations are depicted in Figure 6c. These two lifetime curves had a similar decline trend, which was mainly due to the existing ET processes between Tm^3+^ and Er^3+^ ions (as illustrated in Figure 5). Two double-logarithmic plots of the luminescence intensities of Y_2_O_3_:Er^3+^/Tm^3+^ (8/1 mol%) UCNPs versus the excitation intensity are illustrated in Figure 6d. The slopes of the red (661 nm) and green (564 nm) UCL were 1.6 and 2.0, respectively, indicating that these two UC emissions were ascribed to a two-photon absorption process [50]. All the above results were in good agreement with the assumed transition processes.

### 3.3. Color Tuning under 808 nm Excitation

In order to meet the need of biological applications where the 980 nm excitation source is no longer applicable due to the heavy absorption of water molecules, the UCL properties of the as-prepared UCNPs were evaluated under 808 nm excitation.

Figure 7a,b shows the normalized UC emission spectra of Y_2_O_3_:Er^3+^/Tm^3+^ UCNPs at an excitation intensity of 297 W cm^−2^. For Tm^3+^-free Y_2_O_3_:Er^3+^ UCNPs (Figure 7a), the UCL color changed from green to yellow as the Er^3+^ doping concentrations increased from 0.5 to 8 mol%, and the corresponding R/G ratios enhanced from 0.05 to 1.55. When 1 mol% Tm^3+^ ions were co-doped into the Y_2_O_3_:Er^3+^ UCNPs (Figure 7b), the UCL color can be tuned from yellow to red, and the R/G ratios rose from 0.95 to 6.80. Figure 7c illustrates the comparison of R/G ratios of these two series of UCNPs doped with or without Tm^3+^ ions. Obviously, by introducing 1 mol% Tm^3+^ ions, the R/G ratios have a 4.4-fold enhancement at least, which is similar to the trend under the 980 nm excitation. Additionally, the achieved bright pure red UC emission is much needed for biological applications. Furthermore, we also demonstrated the possible mechanism of ET and BET processes, non-radiative transitions, and UC emissions under 808 nm excitation, as depicted in Figure 7d. Unlike the processes under 980 nm excitation, firstly, the electrons in the ground state of Er^3+^ were excited to the ^4^I_9/2_ state by GSA after absorbing an 808 nm photon, then reached the ^4^I_11/2_ state through a non-radiative transition. After that, the emitting states (^2^H_11/2_, ^4^S_3/2_ and ^4^F_9/2_) could be populated by ETU, CR and BET processes, which were elaborated in Figure 5 [51]. In addition, the power dependence tendencies of the UCL in Y_2_O_3_:Er^3+^ (8 mol%) and Y_2_O_3_:Er^3+^/Tm^3+^ (8/1 mol%) UCNPs are given in Appendix A, which indicates that all the UC emissions also exhibited a two-photon absorption process under 808 nm excitation.

### 3.4. Color Tuning under Pulsed Laser Excitation

Importantly, in addition to the doping concentrations of Er^3+^ and Tm^3+^ ions, the excitation pulse-width will also influence the UC emissions of the UCNPs, which has been reported in previous studies [52,53]. As shown in Figure 8, the R/G ratios of Y_2_O_3_:Er^3+^/Tm^3+^ (8/1 mol%) and Y_2_O_3_:Er^3+^ (8 mol%) UCNPs were examined under the excitation of both a 980 and 808 nm laser with different excitation pulse-widths, and the corresponding detailed spectra are illustrated in Appendix A. In Figure 8a, it can be seen that when the pulse-width of 980 nm laser increased from 0.2 to 4 ms, the R/G ratio of Y_2_O_3_:Er^3+^/Tm^3+^ (8/1 mol%) UCNPs improved greatly from 3.32 to 8.50, indicating that the green UC emission proportion decreased significantly, which is due to the non-steady-state process [52,53]. However, for the Tm^3+^-free UCNPs, the R/G ratio improved slightly from 1.42 to 1.59. Since for Tm^3+^-free UCNPs, the main channels to achieve red emission were the energy transferred from the green-emitting sate by CR1, CR2 and CR3 (Figure 5). The time scales of these processes were short compared to the laser pulse [53], so that the excitation pulse-width basically had no influence on the R/G ratio. However, for Er^3+^/Tm^3+^ co-doped UCNPs, the red-emitting state (^4^F_9/2_) was mainly populated through two fast transition processes (GSA: ^4^I_15/2_→^4^I_11/2_ and ETU: ^4^I_13/2_→^4^F_9/2_) and two slow processes (ET: ^4^I_11/2_→^3^H_5_ and BET: ^3^H_5_→^4^I_13/2_). For the shorter excitation pulse-width, the green-emitting states could still be populated by fast processes (GSA: ^4^I_15/2_→^4^I_11/2_, ESA: ^4^I_11/2_→^4^F_7/2_ and non-radiative transitions: ^4^F_7/2_→^2^H_11/2_ and ^4^S_3/2_), but the red-emitting state (^4^F_9/2_) had not enough time to be populated, which caused the large decline of the R/G ratio.

Additionally, the non-steady-state UCL process was also observed under the excitation of an 808 nm laser, as shown in Figure 8b. For Tm^3+^-free UCNPs, the R/G ratio slightly increased from 1.28 to 1.61, but it enhanced greatly from 2.35 to 6.80 in Er^3+^/Tm^3+^ co-doped UCNPs as the excitation pulse-width increased from 0.2 to 4 ms. The relevant mechanism of this phenomenon is the same as that under 980 nm excitation. Based on the study of the relationship between the R/G ratio and the excitation pulse-width, the ET and BET processes between Er^3+^ and Tm^3+^ ions are further confirmed, and it also offers a guidance for better access to pure red UCL under the excitation of both a 980 and 808 nm pulsed laser.

### 3.5. Ex Vivo Imaging in Biological Tissue

To estimate the potential of the as-prepared Y_2_O_3_:Er^3+^/Tm^3+^ (8/1 mol%) UCNPs in biological applications, its tissue penetration ability was also studied by the ex vivo imaging experiment, as illustrated in Figure 9a. The Y_2_O_3_:Er^3+^/Tm^3+^ (8/1 mol%) UCNPs were dispersed in alcohol solution with a mass concentration of 0.5 mg mL^−1^. Then the solution was injected into a cuboid fresh pork with the size of 6 cm × 2 cm × 2 cm (Appendix A), and the injection points were 1, 3, 5, 7 and 9 mm away from the left edge of the pork, which were marked in red in Appendix A. The excitation laser incident from the left, and the laser power density was 1.2 W cm^−2^ during imaging. The UC emissions were recorded by a CMOS sensor (Samsung ISOCELL HMX, Changsha, China), with the same exposure time of 1 s for each image. A 785 nm short-pass emission filter was applied to prevent the interference of excitation laser to the CMOS sensor.

As shown in Figure 9g,m, there was no UC emission observed from the fresh pork without UCNPs under the excitation of both 808 and 980 nm lasers. After injecting UCNPs, the bright red UC emission from the pork could be observed under the excitation of an 808 nm laser. The UC emission decreased as the tissue thickness increased from 1 to 5 mm (Figure 9b–d). Particularly, when the tissue depths increased larger than 5 mm, the red UC emissions became extremely weak, which were difficult to detect (Figure 9e,f). In comparison, under the excitation of a 980 nm laser, no red UC emission was observed (Figure 9h–l) due to the great absorption of tissue, which makes a 980 nm laser hard to penetrate the biological tissue. The ex vivo imaging experiment shows that these flame-made UCNPs could be detected easily at about 5 mm beneath the tissue surface.

## 4. Conclusions

In conclusion, we successfully prepared Er^3+^-sensitized Y_2_O_3_:Er^3+^/Tm^3+^ UCNPs through the spray flame synthesis method using a swirl-stabilized spray flame reactor with a high-production rate of 40.5 g h^−1^. The fabrication process was continuous, fast, environment-friendly and scalable. After introducing Tm^3+^ ions, the UCL color of the as-synthesized UCNPs could be efficiently tuned from green to red by increasing the Er^3+^ doping concentration from 0.5 to 8 mol% under the excitation of both a 980 nm and 808 nm laser. Here the ^3^H_5_ state of Tm^3+^ ions play the significant role of an intermediate energy state, which help electrons to move from the ^4^I_11/2_ state to the ^4^I_13/2_ state of Er^3+^ ions by the ET and BET processes, and then promote the proportion of red UC emission greatly. A related mechanism was further demonstrated by investigating the lifetimes, the laser power dependence and the excitation pulse-width effect of the UC emissions, which offers a guidance for achieving effective color control in anticounterfeiting and multiplexed labeling applications. Moreover, under the excitation of 808 nm laser, a strong red UC emission was observed in the ex vivo imaging experiment, and the UCNPs can be detected easily at about 5 mm beneath the tissue surface, indicating their great prospects in further biological applications.

## Figures and Tables

**Figure 1 nanomaterials-10-01475-f001:**
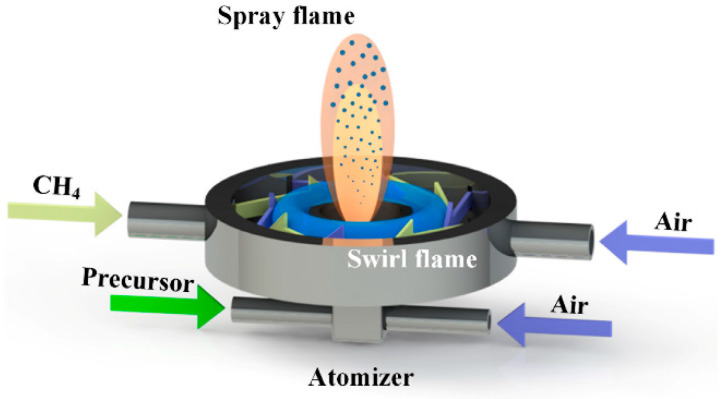
Schematic diagram of the swirl flame assisted gas-phase synthesis method.

**Figure 2 nanomaterials-10-01475-f002:**
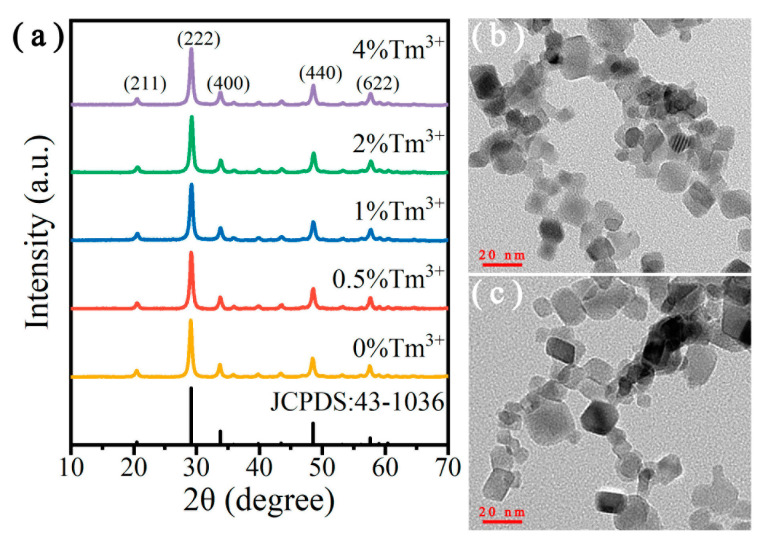
(**a**) XRD patterns of Y_2_O_3_:Er^3+^/Tm^3+^ (8/*x* mol%) upconversion nanoparticles (UCNPs). TEM images of (**b**) Y_2_O_3_:Er^3+^ (8 mol%) and (**c**) Y_2_O_3_:Er^3+^/Tm^3+^ (8/1 mol%) UCNPs.

**Figure 3 nanomaterials-10-01475-f003:**
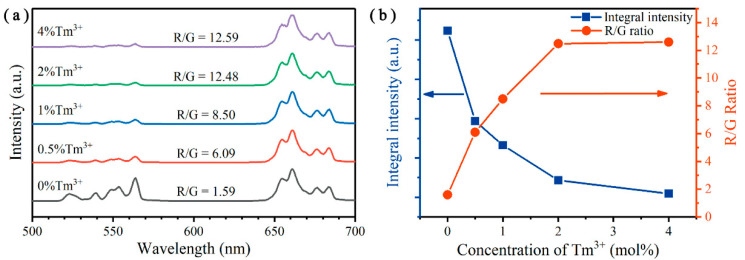
(**a**) Normalized upconversion (UC) emission spectra of Y_2_O_3_:Er^3+^/Tm^3+^ (8/*x* mol%) UCNPs. (**b**) Integral luminescence intensities (from 500 to 700 nm) and red-to-green (R/G) ratios of the Y_2_O_3_:Er^3+^/Tm^3+^ UCNPs doped with different concentrations of Tm^3+^ ions. All excited by a 980 nm laser at an excitation intensity of 341 W cm^−2^.

**Figure 4 nanomaterials-10-01475-f004:**
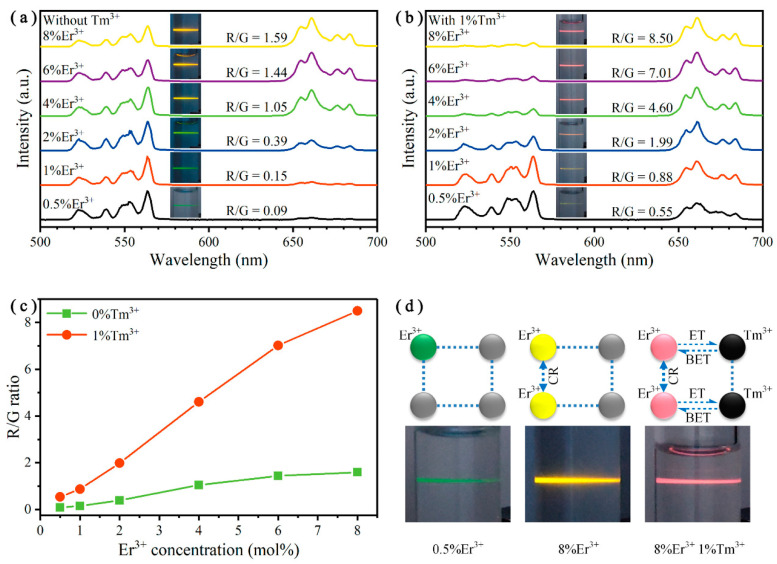
Normalized UC emission spectra of Y_2_O_3_:Er^3+^/Tm^3+^ (*x*/*y* mol%) UCNPs, *x* = 0.5, 1, 2, 4, 6, 8, (**a**) *y* = 0, (**b**) *y* = 1. (**c**) R/G ratios of the Y_2_O_3_:Er^3+^/Tm^3+^ UCNPs doped with different concentrations of Er^3+^ and Tm^3+^ ions. (**d**) Simplified cross-relaxation (CR), energy transfer (ET) and back energy transfer (BET) mechanism. All excited by a 980 nm laser at an excitation intensity of 341 W cm^−2^.

**Figure 5 nanomaterials-10-01475-f005:**
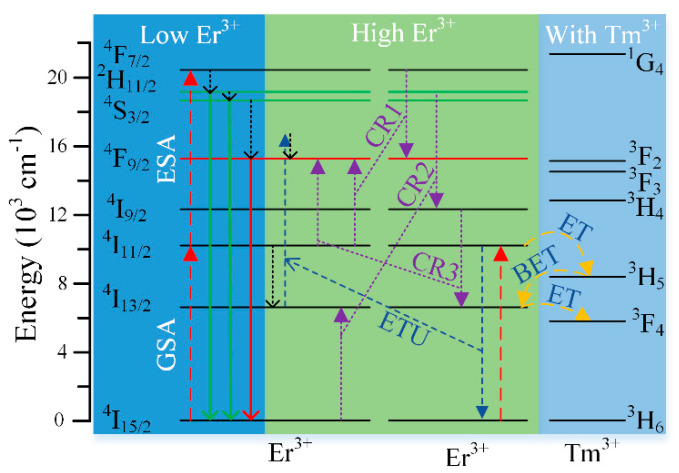
The energy states diagram for Er^3+^ and Tm^3+^ ions under the excitation of the 980 nm laser. The principle of the ET and BET processes, non-radiative transitions, upconversion luminescence (UCL), energy transfer upconversion (ETU) and CR transitions are also presented in the diagram.

**Figure 6 nanomaterials-10-01475-f006:**
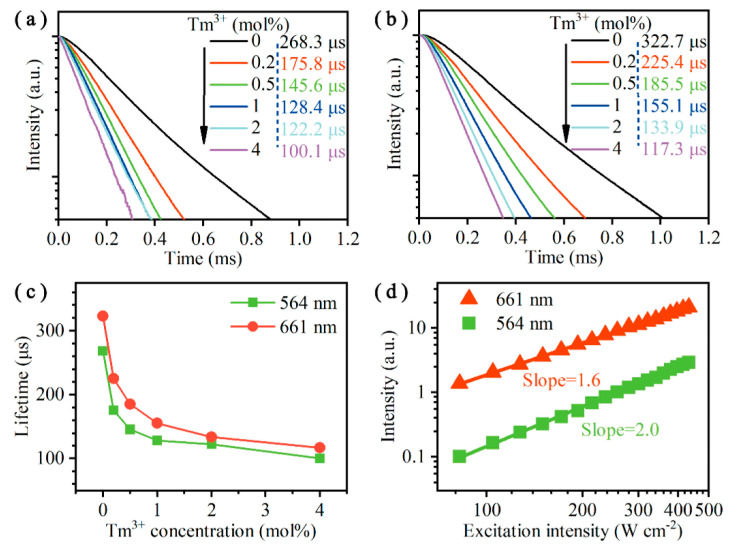
Time–decay curves of the (**a**) 564 nm and (**b**) 661 nm UCL for the UCNPs doped with different concentration of Tm^3+^ ions (the vertical axis is in log scale). (**c**) The lifetime trends of ^4^S_3/2_ and ^4^F_9/2_ states as a function of the Tm^3+^ doping concentrations. (**d**) UCL intensities of Y_2_O_3_:Er^3+^/Tm^3+^ (8/1 mol%) UCNPs as a function of excitation intensity. All excited by a 980 nm laser.

**Figure 7 nanomaterials-10-01475-f007:**
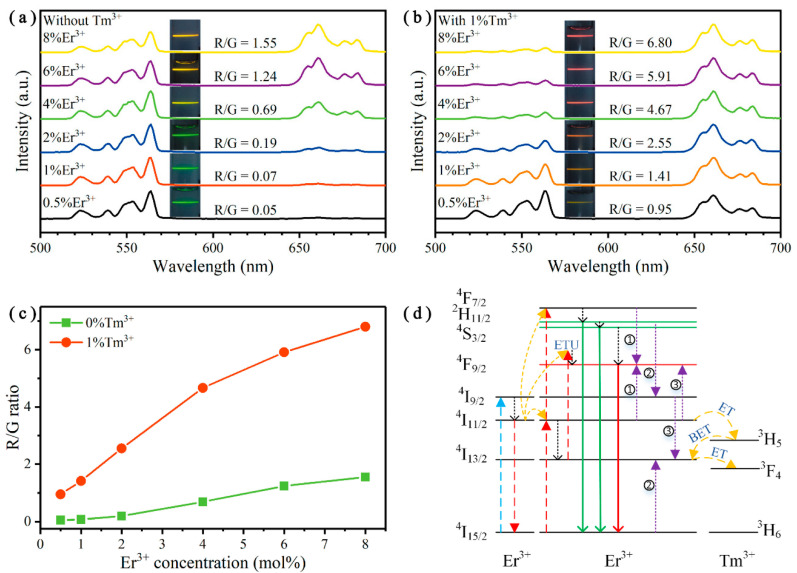
Normalized UC emission spectra of Y_2_O_3_:Er^3+^/Tm^3+^ (*x*/*y* mol%) UCNPs, *x* = 0.5, 1, 2, 4, 6, 8, (**a**) *y* = 0, (**b**) *y* = 1. (**c**) R/G ratios of the Y_2_O_3_:Er^3+^/Tm^3+^ UCNPs doped with different concentrations of Er^3+^ and Tm^3+^ ions. (**d**) ET and BET mechanism between Er^3+^ and Tm^3+^ ions. All excited by an 808 nm laser at an excitation intensity of 297 W cm^−2^.

**Figure 8 nanomaterials-10-01475-f008:**
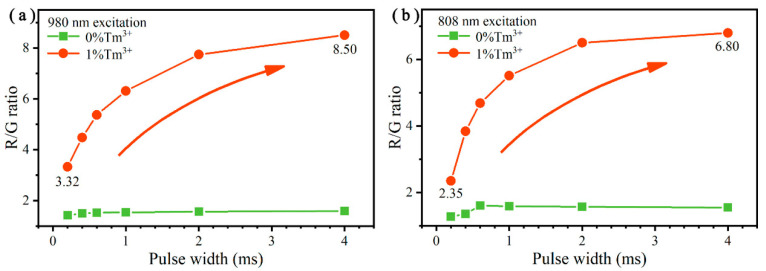
(**a**) R/G ratios of the Y_2_O_3_:Er^3+^/Tm^3+^ (8/*x* mol%) UCNPs under the excitation of a 980 nm laser operated at different excitation pulse-width, *x* = 0 and 1. (**b**) R/G ratios of the Y_2_O_3_:Er^3+^/Tm^3+^ (8/*x* mol%) UCNPs under the excitation of 808 nm laser operated at different excitation pulse-width, *x* = 0 and 1.

**Figure 9 nanomaterials-10-01475-f009:**
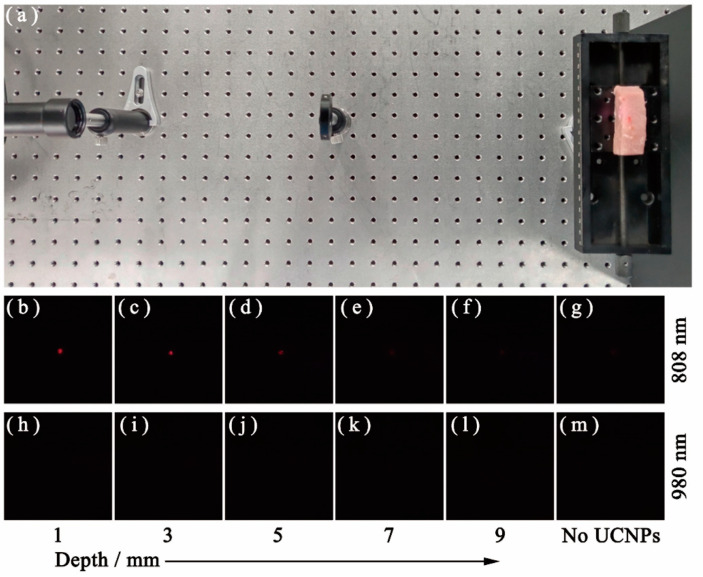
The UC luminescence image of the different depths of pork tissue injected with Y_2_O_3_:Er^3+^/Tm^3+^ (8/1 mol%) UCNPs under the excitation of both 808 and 980 nm laser. (**a**) Experiment setup of the ex vivo imaging experiment. The pork injected with (**b**–**f**) and without (**g**) UCNPs under the excitation of 808 nm laser, respectively. The pork injected with (**h**–**l**) and without (**m**) UCNPs under the excitation of a 980 nm laser, respectively.

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
