# Peer review of "Dual-Wavelength Excited Intense Red Upconversion Luminescence from Er3+-Sensitized Y2O3 Nanocrystals Fabricated by Spray Flame Synthesis"

_nanomaterials, 2020, doi:10.3390/nano10081475_

Round 1

Reviewer 1 Report

Zhao et al. have used spray flame synthesis method to prepare upconverting nanoparticles doped with Er3+ and Tm3+. They report synthesis output of 40.5 grams per hour, which is significant. Other report of color tuning (green to red) under various experimental conditions is also interesting. Others have observed similar effects previously, so this part is less significant. Although the third goal of deep tissue imaging using the developed upconverting nanoparticles is conceptually promising, the measurements shown by the authors are not convincing. It would be great if the authors could quantify the depth upto which they can image rather than crudely showing the photographs that give no quantitative information. 

I recommend publication of the manuscript after revision. There are a number grammatical and linguistic errors that authors should correct. Some of the mistakes (not all) are given below:

Line 37, "Then the investigations ....." remove the word "Then"

Line 56, "efficiently" -> efficient

Line 56, "ET" give the full form of the abbreviation

Line 73, "BET" give the full form of the abbreviation

Line 76 "...  be effective and extensive". Delete "and extensive".

Line 79 " It greatly restricted ...." -> It greatly restricts ...

Line 142 "All the diffraction peaks are corresponding to ..." -> All the diffraction peaks correspond to ...

Line 151 "... morphology which is well agreed with .." -> ... morphology which agrees well with ....

Line 169 "... increases from to 15 mol .." -> ... increases to 15 mol ....

Line 241 "It means that all the CRs can easily occur with the assist of phonons,." -> It means that phonon assisted CRs can easily occur.

Line 264 "... UCL intensities are exponential decay..." -> UCL intensities show exponential decay ...

Author Response

Dear reviewer,

We are glad that our spray flame method for fabricating the upconversion nanoparticles have received your approval, and the color tuning properties under various experimental conditions have met your interest. We appreciate your hard work on providing detailed and valuable suggestions to improve the quality of our work. We are sorry that the third goal of deep tissue imaging is not convincing. According to your suggestion, we have improved the deep tissue imaging experiment to quantify the depth up to which they can image. The details can be seen in the attachment.

We have also carefully read the grammatical and linguistic errors listed in the comment, and corrected the errors sentence by sentence. All the corrections are listed in the attachment. Besides the revisions for grammatical and linguistic errors mentioned above, we have also double-checked the English throughout the manuscript, and all the corrected words are marked in red in the manuscript. We hope the revisions could meet the requirements of publication.

Thank you very much for your help in advance.

Best regards

Reviewer 2 Report

Submitted manuscript presents interesting results and the analysis is related to the presented experimental data. The text is well written. Therefore, in my opinion, this manuscript can be considered for publication in Nanomaterials after minor revision:

1. Energy diagram of Tm3+ ions presented in Fig. 5 should not be simplified as in the manuscript.
2. The emission decay profile should be presented in semi-log scale.
3. Regarding the exvivo experiment: how far from the illuminated side of the meat block were nanoparticles injected? In the picture, it looks like nanoparticles are just under the surface of the block meet. Therefore the title of the subsection “deep penetration in biological tissue is not justified. Additionally, there is no strong difference between the pictures without and with nanoparticles excited with 808 nm while authors mention that bright up-conversion was observed.
4. What was the concentration of the nanoparticles injected into the meat in the ex vivo experiment?

Minor comments:
Line 169: „from” should be removed

Author Response

Dear reviewer,

We are glad that our experimental results have received the approval, and we appreciate your hard work on providing detailed and valuable suggestions to improve the quality of our article. We have carefully revised the manuscript based on your comments and suggestions, and the detailed responses are given in the attachment. We hope the revisions could meet the requirements of publication.

Thank you very much for your help in advance.

Best regards
